# A Retrospective Study of Laparoscopic Cryptorchidectomy in 19 Cats with Intra-Abdominal Testes

**DOI:** 10.3390/ani13010181

**Published:** 2023-01-03

**Authors:** Jesús Villalobos-Gomez, Javier Del-Angel-Caraza, Angelo Tapia-Araya, Fausto Brandao, Carlos Andrés Hernández-López, Franscisco Martínez-Gomariz, Carlos Eduardo Botero-Crespo, Roberto Properzi

**Affiliations:** 1Hospital Veterinario Del Valle, Mexico City 03100, Mexico; 2Sociedad Latinoamericana de Endoscopia Veterinaria-SLEV, Medellin 055420, Colombia; 3Hospital Veterinario para Pequeñas Especies FMVZ-UAEMex, Toluca 50130, Mexico; 4LaparoEndoVet, Mobile Laparoscopy and Endoscopy Service, 43820 Tarragona, Spain; 5Veterinary Minimally Invasive Surgery & Interventional Procedures, 2640-402 Mafra, Portugal; 6Cirugía de Mínima Invasion Veterinaria, Mobile Service, Medellin 055420, Colombia; 7Centro Murciano de Endoscopia Veterinaria, 30150 Murcia, Spain; 8Equipo de Cirugía Minimamente Invasiva Veterinaria, Bogota 11321, Colombia; 9Studio Med Vet Properzi, 16035 Genova, Italy

**Keywords:** cat, companion animals, cryptorchidectomy, cryptorchidism orchiectomy, laparoscopy

## Abstract

**Simple Summary:**

Cryptorchidism is the failure of one or both testes to descend into the scrotum and remain there by 7–8 months of age and is considered the most common congenital disease in male dogs and cats. The prevalence of cryptorchidism in cats is 1.3–6.2%, and can be categorized as unilateral or bilateral and, depending on the location of the testes, as intra-abdominal, inguinal, or pre-scrotal. Due to the risk of testicular torsion, the development of testicular neoplasia and the fact that the cryptorchid testes still produce testosterone, cryptorchidectomy is the treatment of choice for these affected cats. For the intra-abdominal testes, a laparotomy for a caudal midline approach is generally recommended. However, laparoscopy in cryptorchidectomy in cats has been documented in a few clinical cases reports. The aim of the study was to report the short-term clinical outcomes for 19 cryptorchid cats with intra-abdominal testes that underwent cryptorchidectomy with laparoscopic techniques. The results of this study suggest that laparoscopic cryptorchidectomy is an appropriate procedure to treat cryptorchid cats, with all benefits of the minimal invasive surgery, such as a better visibility of abdominal structures, shorter hospitalization times, lower morbidity, less surgical site infections, and most importantly, minimized surgical pain and discomfort.

**Abstract:**

Cryptorchidism is heritable in cats, and due to the pathological risk of testicular torsion, and the development of testicular neoplasia, cryptorchidectomy is the treatment of choice. For the intra-abdominal testes, a laparotomic approach is generally recommended; however, laparoscopic cryptorchidectomy in cats has been documented in a few clinical case reports. The aim of the study was to report the short-term clinical outcomes for 19 cryptorchid cats with intra-abdominal testes that underwent cryptorchidectomy with laparoscopic techniques. Medical records of client-owned sexually intact male cats that underwent laparoscopic cryptorchidectomy in different veterinary hospitals or ambulatory surgical services were reviewed. The procedure was performed in 19 cats. The average time (mean ± standard deviation, SD) for all procedures was 23 ± 6 min (range 15–35 min). The time for laparoscopic removal of a single abdominal testis was 22 ± 6 min, with 30 min for the bilateral abdominal testes. The time until hospital discharge varied depending on the surgeon’s criteria, with a mean of 6 h (range 3–24 hrs). None of the cases analyzed showed any surgical complications. The results of this study suggest that laparoscopic cryptorchidectomy is an appropriate surgical procedure to treat cryptorchid cats with intra-abdominal testes, all with benefits of minimal invasion surgery.

## 1. Introduction

Normally, the testes descend into the scrotum shortly after birth; however, they may move freely up and down in the inguinal canal prior to puberty. Cryptorchidism is the failure of one or both testes to descend into the scrotum and remain there until 7–8 months of age, and is considered the most common congenital disease in male dogs and cats [1,2].

Cryptorchidism is considered heritable in cats, and a polygenic mode of inheritance has been suggested [1,2]. The prevalence of cryptorchidism in mixed breed or purebred cats is 1.3–6.2%, with possibly the highest incidence in the Persian and Ragdoll breeds [1,3,4,5,6].

Cryptorchidism can be categorized as unilateral or bilateral and depending on the location of the testis, as intra-abdominal, inguinal, or pre-scrotal. Most cryptorchids are unilateral (78–90%), with left and right sides being equally affected [1]. In the study of Richardson and Mullen, the location of the retained testes in cats was 49% inguinal, 33% intra-abdominal, and 14% pre-scrotal [7]. Cats with bilateral cryptorchidism often have intra-abdominal testicles; in these cases, it is essential to check the presence of testosterone-dependent penile spines, which should be atrophied six weeks after complete castration [1]. The diagnostic evaluation must be complemented with abdominal or inguinal ultrasonography [8]; due to the risk of testicular torsion, the development of testicular neoplasia and the fact that the cryptorchid testes still produce testosterone, cryptorchidectomy is the treatment of choice for these affected cats [2]. According to the literature, a laparotomy in the caudal midline approach is generally recommended to remove the intra-abdominal testes [3,7]. However, the laparoscopic, and laparoscopic-assisted cryptorchidectomy in cats has been documented in a few clinical case reports [2,9,10,11,12].

The objective of the study was to report the short-term clinical outcomes for 19 cryptorchid cats with intra-abdominal testes that underwent a cryptorchidectomy with a laparoscopic technique.

## 2. Materials and Methods

Medical record reviews of client-owned sexually intact male cats that underwent laparoscopic cryptorchidectomy in different veterinary hospitals or mobile surgical services were obtained from members of the Sociedad Latinoamericana de Endoscopia Veterinaria-SLEV. Data on breed, age, weight, and body condition score assigned based on a standardized scoring system ranging from 1 to 9 [13], as well as categorization of the kind of cryptorchidism as unilateral right or left or bilateral were recorded.

The intraoperative data included the technique for trocar insertion, location of trocar placement, and type of procedure (laparoscopic or laparoscopic-assisted), the type of haemostasias and cutting technique; time of surgical procedure (defined as the time from the initial incision of the first trocar, to the last incision closure); the occurrence of possible surgical complications as a trans-surgical complications (bleeding, tissue injury, or need to convert to an open procedure); immediate post-surgical complications (during hospitalization, as including an bleeding or pain), or later post-surgical complications (8–10 days after surgery, such as surgical wound infection or pain); the clinical follow-up of the patients was carried out by telephone with the help of the referring doctors.

## 3. Results

The clinical cases registered in different veterinary surgical centers of Latin America (Mexico and Colombia) and Europe (Portugal, Spain, and Italy) were reviewed, and the records of 19 cryptorchid cats with intra-abdominal testes were found. Breeds represented consisted of 14 domestic shorthair, two Siamese, and one each of Bengal, Persian and Sphynx cats.

The mean age of cats was 2.5 years (ranging from 9 months to 7 years), with 63.2% in the junior life stage (7 months to 2 years) and 36.8% in the prime life stage (3–6 years).

Mean weight of the cats was 4.2 kg (ranging from 3.0 to 5.5 kg). Based on the body condition score (9/9 scale), 12 cats had an ideal weight (5/9), 63.1%, five were overweight (7–8/9), 26.3%, and two were underweight (3/9), 10.6%.

Of the 19 cryptorchid cats with intra-abdominal testes studied, 11 were right unilaterally cryptorchid (57.9%), five left unilaterally cryptorchid (26.3%), and three bilaterally cryptorchid (15.8%).

For the surgical procedure, in all cases, inhaled anesthesia was used, and anesthesia monitoring was performed by qualified personnel. In all cases, the main trocar was placed just caudal to the umbilicus using a Hasson technique in 10 cats, modified-Hansson in six cats, and the Veress needle technique in three cats. A small skin incision (<10 mm) was made at this site, and subcutaneous tissues were dissected to visualize the external rectus abdominis sheath; a small stab incision was made into the peritoneal cavity. In 16 cats, a 5 mm trocar was inserted through the incision, and a 6 mm trocar into the remaining 3 cats. The capnoperitoneum was achieved at a rate of 1–2 L/min to an intrabdominal pressure of 7–9 mmHg. A rigid 5 mm telescope with 0° or 30° angle vision was inserted through the trocar, and the abdomen was explored systematically. When an abdominally retained testes was detected, a second access port with a 5 or 6 mm trocar was inserted in a caudal position and slightly lateral to the retained testis, and particularly in the case of the three bilaterally cryptorchid cats, the second trocar was inserted in the center of the caudal abdomen. The second trocar placement was performed with the use of direct visualization through the laparoscope, and transillumination of the abdominal wall was used to avoid blood vessels. To be able to work with two trocars, external suture was used to suspend the testis. The size of the needle was dependent on the size of the animal but had a curvature and sufficient length to allow comfortable entry and exit through the abdominal wall. In the current clinical cases, a 3/8 circle reverse cutting needle ranging from 24 to 30 mm length was commonly used. With transillumination, the most appropriate place to insert the needle was located, avoiding vascularized areas. First, the needle was passed through the abdominal wall, under direct endoscopic visualization and then passed through the testis carefully avoiding vascular structures, and ultimately pulling the needle out through the abdominal wall again. The suture was secured externally with the help of a hemostat, thus fixing the testis to the abdominal wall. In three cases, a percutaneous grasping technique was performed using an open-loop grasper that allowed the testis to be fixed to the abdominal wall with a single puncture; and in three cases, a third port was implemented with two paramedian trocars. With these maneuvers, the detailed visualization of the vascular pedicle, the spermatic cord, and the gubernaculum was achieved (Figure 1 and Figure 2).

In 13 cases, the coagulation and cutting of the vascular pedicle, the spermatic cord, and the gubernaculum were performed with advanced electrothermal bipolar energy devices (eight with the Laparoscopic Tissue Sealer G2-ENSEAL^®^ Ethicon-Johnson & Johnson (New York, NY, USA) and five with LigaSure^TM^ Covidien-Medtronic (Dublin, Ireland), https://www.jnjmedtech.com/global, https://www.medtronic.com/covidien/en-us/index.html accessed on 27 October 2022) (Figure 3). In five cases, this procedure was made with a Harmonic scalpel device (Figure 4 and Appendix A) (Harmonic ACE^®^ +Shears and Gen11 generator Ethicon-Johnson & Johnson), and in one case, with the use of a monopolar energy hook.

In 18 cases, it was possible to achieve the complete removal of the testis, without evidence of bleeding. Following complete resection, the caudal trocar was removed, and the trocar incision was widened sufficiently (<10 mm) for exteriorization to extraction of the testes from the abdomen. In one case of bilateral cryptorchidism, a laparoscopic-assisted procedure was performed; the testes were exteriorized, extracorporeal dissection of the gubernaculum, and a transfixation ligature suture (3–0 absorbable monofilament suture) of the vascular pedicle and spermatic cord was applied. After the testes excision, for the remaining anatomical structures, vascular stumps were reintroduced into the abdominal cavity. In all cases, after removal of the cryptorchid testes, routine exploratory laparoscopy was conducted in order to assess any bleeding or tissue trauma; excessive CO_2_ was removed, and each trocar site portal was routinely closed with a 3–0 absorbable monofilament suture.

For 18 cases, full laparoscopic cryptorchidectomy was achieved, and for only one case, the laparoscopic-assisted technique was selected. The average time (mean ± SD) for full laparoscopic procedures was 23 ± 6 min (range 15–35 min). The time for the laparoscopic removal of a single abdominal testis was 22 ± 6 min, and 30 min were needed for the bilateral abdominal testes. The time until hospital discharge varied depending on the surgeon’s criteria, with a mean of 6 h (range 3–24 h). In all cases, a rutinary post-surgical analgesia and antibiotic management with conventional drugs was performed for a few days. None of the cases evaluated demonstrated any perioperative complications.

## 4. Discussion

During routine physical exams, testes inspection and palpation should be conducted as a standard practice for all male cats. Cryptorchidism diagnosis is achieved by careful examination, checking for the absence of one or both testes within the scrotum. In one study, only 22% of the cat owners knew their cat was cryptorchid [5]. In our studied cases, 63.1% were cats older than 2 years, which indicates a late diagnosis of this condition; it is generally accepted that this is because both testes are fully descended into the scrotum and remain there by 7–8 months of age [1]. The breed, weight and body condition scores did not have any relevant association in cryptorchid cats.

Of the cryptorchid cats with intra-abdominal testes studied here, the most common configuration was unilaterally cryptorchid, with 84.2%, affecting mainly the right testis, and bilaterally cryptorchid, with 15.8%. Similar results have been described in cryptorchid dogs [4,5,11]. In this species, this alteration has been associated with the cranial position of the right testicle in the abdomen, which is normally more cranial than the left; this is reflected in a slower descent of the right testicle since the distance to be covered to reach the scrotum is longer [4].

Traditionally, intra-abdominal testes were removed through a conventional laparotomy or minimal laparotomies with the use of a spay hook, with potential complications such as trauma of the abdominal structures [11]. Usually, a caudal midline laparotomy is required to find the testes, either by going from the testicular artery, starting at the level of the fourth lumbar vertebra, or by following the deferent duct from the prostate or the gubernaculum from the inguinal ring [2]. The location of the intra-abdominal testes may vary depending on the position of the patient or bowel peristalsis, as they are much more mobile than an ovary; therefore, it is not recommended to use a spay hook to find a testis as this may increase the risk of ureteral and urethral trauma [2,11]. In conventional laparotomy, longer healing times and a higher risk of hernias and eviscerations are likely. Laparoscopy offers important advantages in cats with cryptorchidism. The two small holes needed for laparoscopy (<10 mm), even in cases of bilateral cryptorchidism, are much less painful than a large midline laparotomy and offer considerably better visualization, especially for the inguinal region.

In the literature, the reported laparoscopic procedures for the cryptorchidectomy in cats refer to the laparoscopic-assisted technique in five cats [2,11,12]; only for one cat, a complete laparoscopic procedure with three ports was performed [9]; and in three cats, a single-port devise was used [10]. In our study, 18 procedures (95%) were performed with the completely laparoscopic procedure through two or three ports and with the use of advanced bipolar energy devices and only one with the laparoscopic-assisted technique. Surgical time and the time until hospital discharge was similar to those reported in other studies [10,11].

According to Miller et al. (2004), the general potential complications for laparoscopy include cardiovascular and pulmonary changes associated with carbon dioxide capnoperitoneum; trocar injuries to the urinary bladder, small and large intestines, or major vessels; subcutaneous emphysema; visceral herniation; wound infection or hematoma formation at the trocar entry sites [11]; however, they have mainly been described for humans. In veterinary medicine, differences in results and complications associated between laparoscopic and laparotomy cryptorchidectomy have not been investigated. Nevertheless, the benefits of laparoscopic procedures are undeniable. These include better visibility of abdominal structures, lower perioperative morbidity, fewer surgical site infections and, most importantly, less pain [2]. In our study, the time until hospital discharge of the patient on average was 6 h, and none of the clinical cases studied showed any perioperative complications. Unfortunately, based on the study design, and because the patients were only referred for a minimally invasive surgical procedure, it was not possible to obtain more information regarding cryptorchidism in each of the clinical cases; this is the main limitation of this study.

## 5. Conclusions

The results of this multicentric study suggest that the laparoscopic or laparoscopic-assisted cryptorchidectomy with two ports is an appropriate surgical procedure to treat cryptorchid cats with intra-abdominal testes, with all benefits of minimal invasion surgery.

## Figures and Tables

**Figure 1 animals-13-00181-f001:**
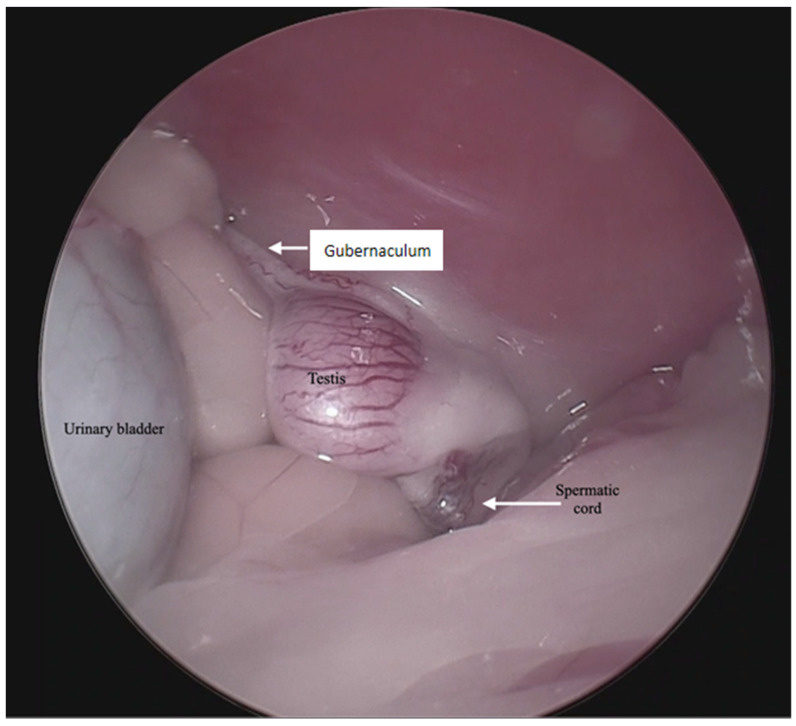
Endoscopic image of the aspect of an intra-abdominal testis.

**Figure 2 animals-13-00181-f002:**
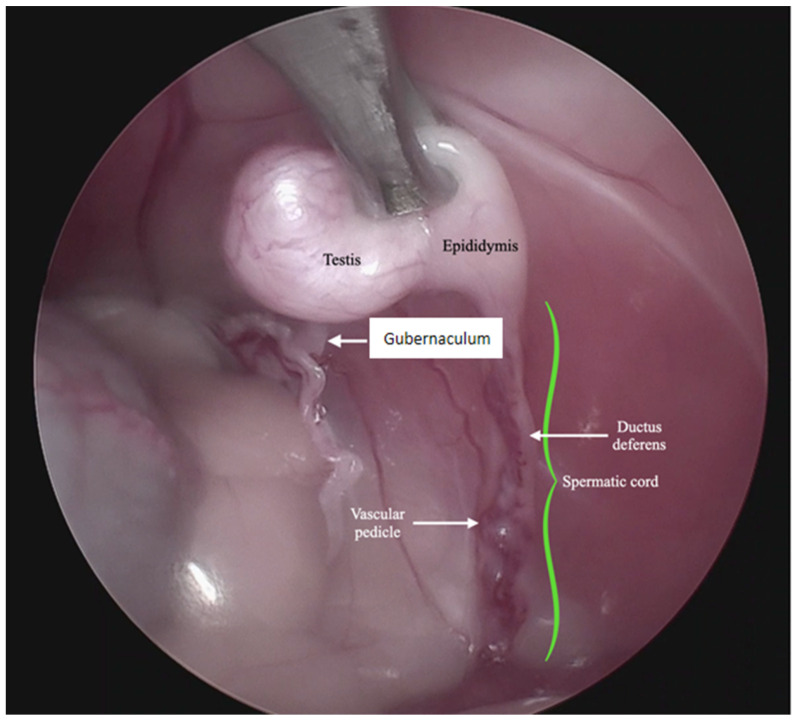
Endoscopic image of the anatomical structures in the intra-abdominal testis.

**Figure 3 animals-13-00181-f003:**
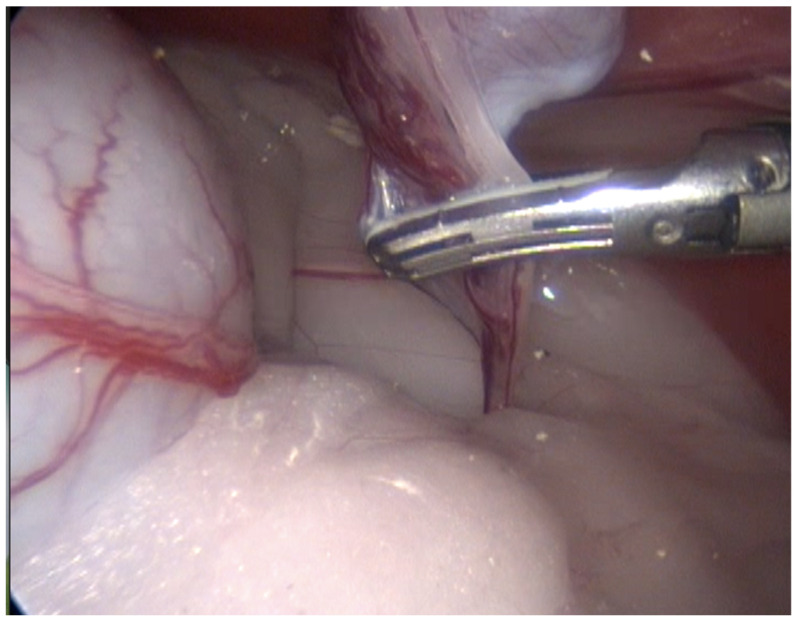
Technique with two access ports. View of the coagulation and cutting maneuvers of the vascular pedicle, spermatic cord, and gubernaculum, performed with an advanced bipolar energy device.

**Figure 4 animals-13-00181-f004:**
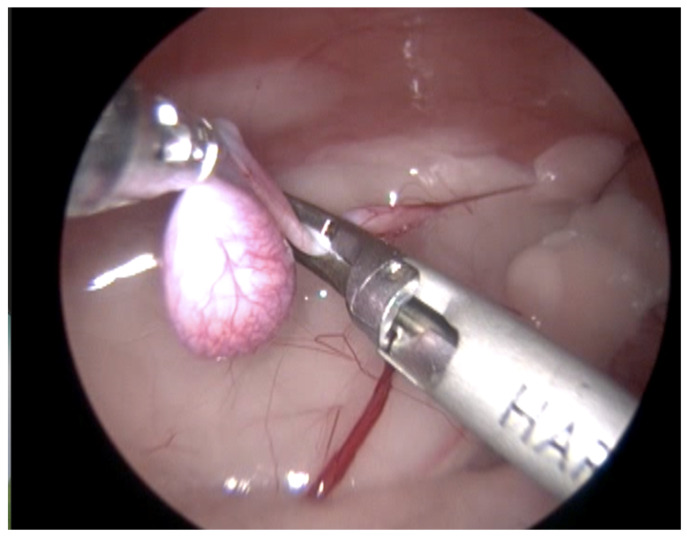
Technique with three access ports. View of the maneuvers of the testis support with a grasping of forceps and coagulation and cutting of the vascular pedicle, spermatic cord, and gubernaculum, performed with a harmonic scalpel device. (See Appendix A.)

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
