# Peer review of "A Retrospective Study of Laparoscopic Cryptorchidectomy in 19 Cats with Intra-Abdominal Testes"

_animals, 2023, doi:10.3390/ani13010181_

Round 1

Reviewer 1 Report

The work is a case contribution on the laparoscopic and laparoscopic-assisted surgery for abdominal testis removal in unilateral or bilateral cryptorchid cat. The author’s intention is interesting because the laparoscopic cryptorchidectomy in cat is rarely performed and the reports on this topic are rather limited.

In the introduction, the authors adequately describe epidemiological data, preoperative diagnosis and surgical criteria commonly adopted for cryptorchid cats.

In the materials and methods section, the data obtained from the clinical case history and the laparoscopic surgical criteria adopted are adequately detailed. However, since one of the main justifications for the proposed technique is represented by the intra- and post-operative pain reduction, the follow-up evaluation criteria should be described in more detail.

The results are well described, and the iconography and the supplementary video file adequately describe the surgical work done. Follow-up data should be reported.

The discussion I suggest to better describe the outcomes of the different surgery techniques, as the authors affirm that the laparoscopic technique is better for patient recovery than laparotomy This would add value to the paper and would represent a useful guide to address the surgeon’s choice.

 Introduction. Line 53. Change ‘feral’ to ‘mixed bred’

 Results. Lines 106-109. This paragraph should be moved in the Material and Methods Section. In fact, these data do not add anything to the results and are not discussed.

Line 130: change ‘deepened’ to ‘depended’

Lines 143-144. ‘In 13 cases, the coagulation and cutting maneuver of the vascular pedicle, the spermatic cord and the gubernaculum were performed.’ Please check the English of this sentence.

 Discussion. The discussion section is not exhaustive. The benefits of the proposed surgical technique should be based on a more rigorous comparison with literature data.

Lines 232-236. ‘Nevertheless, the benefits of laparoscopic procedures are undeniable, such as a better visibility of abdominal structures, shorter hospitalization times, lower morbidity, less surgical site infections and, most importantly, less pain [2]. In our study, the time until hospital discharge of the patient on average was 6 hours, and none of the clinical cases studied showed any trans-surgical or post-surgical complications’.

A more rigorous comparison of hospitalization times, morbidity, surgical site infections and pain with literature data of classical laparotomic surgical cryptorchidectomy should be provided.

Bibliography. The bibliography is relevant and correctly reported. If available, papers containing hospitalization times, morbidity, surgical site infections and pain data from laparotomic surgical cryptorchidectomy.

Author Response

Dear reviewer, we appreciate your objective comments and suggestions, all of them were analyzed, and the respective corrections were made to the manuscript.

Reviewer 2 Report

The results of this work show an interesting contribution to the surgical approach in cryptorchid cats.

In the Introduction you could specify the benefits of the laparoscopic technique. 

It would also be useful to describe the type of anaesthesia and analgesia used and the timing.  To this end, it should be specified whether the retained testicles have been ultrasound visualised beforehand, as this can reduce the surgical execution time.

 This would allow us to better understand the benefits of the technique you used.  

It is advisable to review the manuscript for the English language

Author Response

(The authors gave the same response as above.)

Reviewer 3 Report

Dear Authors, 

the submitted study is interesting yet the discussion should be improved in order to address the advantaged of laparoscopic surgery compared to the laparotomic approach. Although you repeatedly name shorter hospitalization time, less infection etc as advantages, you fail to describe incidences of these factors in laparotomic surgeries. Please adjust and provide a comparison of these factors between your results and the literature regarding laparotomic criptorchidecdomy within the discussion.
Some other, small changes may be necessary as mentioned here:

Line 23: laparoscopy instead of laparoscopic

Line 27: invasive instead of invasion

Line 33: lapartomic instead of laparotomy

Line 36: verb is missing

Line 44: order alphabetically

Line 48: “however they may move”

Line 51-52: insert citation

Line 59: “Bilateral cryptorchid cats…” – delete In the

Line 60-61 not clear – rephrase

Line 66: using instead of for

Line 70: delete “some”

Line 84: delete “as the main limitation of this study” mention it in discussion

line 114: and anesthesia monitoring was performed by qualified personnel

line 125: ventral is not the correct word as you are working on the abdomen: caudally?

Line 134-line 137: rephrase

Line 140: which allowed fixation of the testis to the abdominal wall…

Line 141 “-two laterals”- rephrase or explain better

Line 171: removal or detachment instead of release?

Line 188 postoperative antibiotic treatment was due to laparoscopic approach? Is it a general routine in all surgical centers? Should be mentioned in discussion

Line 219: delete “and”

 Line 231: in veterinary medicine

Line 234 please compare hospitalization times between laparotomic approach and laparoscopic approach. As far as I am aware, this type of procedure is performed in day hospital even if a laparotomic approach is performed. Therefore, a comparison within the discussion should be made or if such a comparison is not possible, “shorter hospitalization time” should be removed from the aim as you have no means to evaluate this

Line 235-237: same as for line 234 – please insert studies and/or case reports which give an idea on the incidence and type of trans-surgical or post-surgical complications in a laparotomic approach

Author Response

(The authors gave the same response as above.)

Round 2

Reviewer 2 Report

this article may be published in Animals